# Difficult and Severe Asthma in Children

**DOI:** 10.3390/children7120286

**Published:** 2020-12-10

**Authors:** Federica Porcaro, Nicola Ullmann, Annalisa Allegorico, Antonio Di Marco, Renato Cutrera

**Affiliations:** Pediatric Pulmonology & Respiratory Intermediate Care Unit, Sleep and Long-Term Ventilation Unit, Academic Department of Pediatrics, and Genetic Research Area, Bambino Gesù Children’s Hospital, IRCCS, 00165 Rome, Italy; nicola.ullmann@opbg.net (N.U.); annalisa.allegorico@opbg.net (A.A.); antonio.dimarco@opbg.net (A.D.M.); renato.cutrera@opbg.net (R.C.)

**Keywords:** asthma, difficult asthma, severe asthma, children, biologic drugs

## Abstract

Asthma is the most frequent chronic inflammatory disease of the lower airways affecting children, and it can still be considered a challenge for pediatricians. Although most asthmatic patients are symptom-free with standard treatments, a small percentage of them suffer from uncontrolled persistent asthma. In these children, a multidisciplinary systematic assessment, including comorbidities, treatment-related issues, environmental exposures, and psychosocial factors is needed. The identification of modifiable factors is important to differentiate children with difficult asthma from those with true severe therapy-resistant asthma. Early intervention on modifiable factors for children with difficult asthma allows for better control of asthma without the need for invasive investigation and further escalation of treatment. Otherwise, addressing a correct diagnosis of true severe therapy-resistant asthma avoids diagnostic and therapeutic delays, allowing patients to benefit from using new and advanced biological therapies.

## 1. Introduction

Asthma is the most frequent chronic inflammatory disease of the lower airways that affects 4.4% of preschoolers and 6.4% of primary school children [1] and is responsible for significant social and economic burdens [2].

Good control of symptoms is usually achieved with low doses of inhaled corticosteroids (ICS), and only a small proportion of asthmatics need high doses of ICS, long-acting β2-agonists (LABA), and leukotriene antagonists to achieve symptom control [3]. In 2008, the concept of “problematic severe asthma” was suggested to indicate children not responding to high doses of asthma medications [4]. Ten years later, this concept was revised as it was considered reductive, being based solely on prescribed treatment levels [5]. Indeed, there might be multiple reasons for poor control of asthma symptoms in children, such as (1) incorrect diagnosis; (2) asthma associated with other comorbidities; (3) difficult asthma, and (4) severe asthma [5].

As the management of children with poor control of asthma symptoms still represents a clinical challenge to pediatricians, we carried out a literature review to provide an overview of the definition of *difficult* and *severe asthma* and their respective strategies of treatment.

## 2. Is It Asthma?

In patients with a history of therapy-resistant asthma, the first step is to go back to the basics and confirm the initial diagnosis. We need to recollect a detailed clinical history, perform a complete physical examination, and respiratory functional tests [6].

First of all, the diagnosis of asthma is supported by the presence of (1) chronic airway inflammation triggered by exposure to respiratory viruses or specific allergens in sensitized children, and (2) airflow limitation that is completely or partially reversible following the administration of inhaler short-acting β-2-agonists (SABA) [7,8].

Parentally reported symptoms raising the suspicion of asthma include episodes of shortness of breath, wheezing, chest tightness, and periodic nocturnal dry cough. However, a symptom-based approach diagnosis can be misleading, resulting in over and under-diagnosis of asthma [9,10,11], and consequently in unnecessary treatment, increased disease burden, and significant impact on the patients’ quality of life.

The detection of wheezing on clinical examination reinforces the suspicion of asthma, even though a negative physical examination cannot exclude it. Based on reported symptoms, physicians should consider performing diagnostic tests, such as spirometry with or without reversibility testing, fraction of exhaled nitric oxide (FeNO) measurements, and peak expiratory flow (PEF) variability over 2–4 weeks to confirm the diagnosis of asthma [12].

Spirometry with bronchoreversibility is the first line objective test recommended in most pediatric asthma guidelines for school-age children. Reversible airflow obstruction characterized by FEV_1_ improvement of ≥ 12% is the hallmark of asthma [13].

FeNO measurement is suggested to support the diagnosis of asthma, as well as to define disease severity and adherence to prescribed therapy [12]. In children, a FeNO level ≥35 ppb should be interpreted as an index of bronchial inflammation [12].

PEF measurement is the maximum airflow (L/min) during a forced expiration beginning with fully inflated lungs. Variations of 20% of PEF over 2–4 weeks observation period, with twice-daily measurements, are suggestive of asthma [14].

Unfortunately, the diagnosis of asthma in preschool children is often based on history and treatment response because of the inability to perform the above-mentioned lung function tests. Although plethysmography, interrupter technique (Rint), and forced oscillation technique (FOT) can be performed to measure respiratory resistance (Rrs) and reactance (Xrs) in children aged >2 years, its use is limited in routine clinical practice because these techniques are performed in a few specialized centers [15]. Rrs measures the degree of obstruction of the main central airways, while small airways (<2 mm in diameter) account for only 10% of total airway resistance. Xrs measures the elastic recoil forces of the respiratory system and determines the effective ventilation of the lung’s peripheral areas. Although the scientific literature is conflicting, these parameters appear to be an important tool for the assessment of bronchial asthma in children [16].

## 3. Is It a Real Therapy-Resistant Asthma?

The typical and most frequently reported symptoms of asthma, dry cough, shortness of breath, wheezing, dyspnea on exertion, and chest tightness are shared with many other disorders that can be confused with asthma or which may complicate asthma. Therefore, patients with asthma symptoms that are refractory to traditional asthma treatment should be evaluated for alternative conditions that can mimic asthma. A symptoms-guided diagnostic workup should be considered in patients with an unclear history of asthma and/or suboptimal response to standard asthma treatment.

For instance, the presence of stridor at rest or stridor on exertion, wet cough, barky cough, failure to thrive, heartburn, and drumstick fingers require a careful evaluation for additional investigation (Table 1) [17,18].

## 4. Difficult Asthma

Difficult asthma is asthma that is uncontrolled despite GINA step 4 and 5 (medium or high dose ICS with a second controller; maintenance of oral corticosteroids), or that requires such treatment to maintain good symptoms control and reduce the risk of exacerbations. In this case, the lack of symptom control is linked to the presence of comorbidities or poor adherence to medical prescriptions. Accordingly, once a diagnosis has been confirmed, all possible risk factors or comorbidities need to be considered in patients with persistent symptoms despite standard treatment [19,20].

Periodical and careful assessments carried out by the specialist and by the family pediatrician can help identify potentially modifiable factors responsible for poorly controlled asthma. Indeed, poor symptom control is a consequence of modifiable factors that need to be carefully assessed, such as (1) nonadherence to medication or inadequate inhalation technique, (2) persistent environmental exposures, (3) comorbidities, and (4) psychosocial factors.

### 4.1. Adherence to Medication

Good adherence is most commonly defined as taking between 70–80% of prescribed treatment [21]. Suboptimal adherence to ICS leads to poor asthma control, severe asthma attacks, and frequent use of healthcare resources [22]. Although it is known that half of difficult-to-treat patients have poor adherence to prescribed medication or improperly use the suggested devices [23,24], clinicians are not used to always check adherence and inhaler technique at the time of patient’s evaluation [25]. Explaining and showing the spacer’s correct use is the most effective model to improve the inhaler technique and asthma control [26,27].

Several measurement tools, both subjective and objective, have been developed to assess adherence in adults and children with asthma [28] (Table 2). Unfortunately, each of these measures has limitations, such as the unavailability of self-report adherence questionnaires validated for the pediatric population, the often poor availability of electronic monitoring devices (EMD), the high production costs, the ability of patient/parent to manipulate the data, and the ability of EMD to measure inhalation and inhaler technique [28].

### 4.2. Environmental Exposures

#### 4.2.1. Tobacco Smoke

It is known that exposure to environmental tobacco smoke increases pediatric asthma severity [29] and, in particular, increases resistance to steroids [30]. Therefore, the assessment of passive and/or active smoke exposure is mandatory for all children with difficult asthma. As parents and patients (especially if teenagers) can deny the exposure to tobacco smoke, levels of salivary or urinary cotinine can be used to determine actual exposure [31].

#### 4.2.2. Air Pollution

There is increasing evidence of the association between air pollution and asthma exacerbations as well as new onset of asthma in children. Air pollution is a mixture of particles and gases emitted from several sources or generated in the atmosphere through chemical reactions (fine particles < 2.5 µm in diameter, nitrogen dioxide, and ozone). All these elements can cause oxidative stress into the airways, leading to inflammation and remodeling, especially in asthmatic children [32].

#### 4.2.3. Allergen Exposure

Children with poor asthma control despite proper treatment should be investigated for allergy sensitization [33]. Several studies reported the increased risk of asthma in children with a family history of atopy, early-onset atopic dermatitis, sensitization to egg or milk in the first years of life [34]. In addition, the poor control of asthma symptoms is directly correlated with both specific IgE levels and the number of sensitizations [33]. Consequently, it is essential that allergen exposure is minimized in all patients with difficult asthma before any drug escalation.

### 4.3. Comorbidities

Co-morbidities are important in the management of difficult and severe asthma [18], as they may contribute to poor disease control, as well as mimicking asthma symptoms (Table 3). Researching and optimizing the management of these conditions also through a multidisciplinary team is mandatory in all asthmatic patients with poor symptom control [35].

### 4.4. Psychosocial Factors

Although the literature on psychiatric and behavioral disorders among children with asthma is conflicting, most research reported that children with asthma display more emotional and behavioral problems than their healthy peers [36].

Anxiety, depression, and symptoms of inattention, hyperactivity, and oppositional behaviors are often reported by patients and their parents. Children with asthma and internalizing disorders are at risk of having worse asthma control, increased use of rescue medications, more access to healthcare facilities for attacks, poorer pulmonary outcomes, and more missed school days [37,38,39]. Moreover, the caregivers of children with asthma can suffer too from chronicity, developing emotional difficulties that can interfere with the management of the young patient.

Questionnaires assessing the quality of life for both the child and family (Pediatric asthma quality of life questionnaire; PAQLQ) [40] as well as symptom control (Asthma control test; ACT) [41] are useful tools to estimate the severity and the impact of the disease on patient’s life. Consequently, psychosocial interventions, including educational programs, behavioral support, cognitive-behavioral therapy, and family interventions can be considered to reduce the psychological impact of the disease and to better control asthma symptoms.

### 4.5. Socioeconomic Factors

In addition, we must not forget that low socioeconomic status (low income, educational level, parents’ occupation) is often associated with poor asthma control (need for rescue therapy for asthma exacerbation, need for emergency health service visits, need for hospitalization for asthma exacerbation and fatal outcome) [42]. In these cases, the role of the family pediatrician becomes fundamental in identifying children vulnerable to asthma with a worse prognosis.

## 5. Severe Asthma

Severe therapy-resistant asthma is defined by the need for high dose ICS (Table 4 and Table 5), plus a second controller (long-acting β2-agonist or a leukotriene antagonist) for the previous year or systemic corticosteroids for at least 50% of the previous year, to prevent asthma from becoming “uncontrolled” or remaining “uncontrolled” despite this therapy [43]. These criteria must be met in patients in which comorbidities or modifiable factors have been correctly managed.

Patients affected by severe asthma experience frequent exacerbations and might suffer from iatrogenic damage related to corticosteroids use (obesity, diabetes, hypertension, depression) [44]. All these conditions impact a patient’s quality of life and interfere with family and social life [45]. Since only a small percentage of asthmatic patients is affected by severe asthma (up to 5% of all children with asthma) [46], this clinical entity is still poorly known and, therefore, represents still a challenge for physicians.

All children meeting the criteria for severe asthma should be referred to a tertiary pediatric respiratory center for a full multidisciplinary assessment that includes several investigations defining the clinical phenotype and inflammatory endotype in order to better optimize the management [47].

Asthma phenotype can be classified according to the age of symptoms onset, allergen sensitization, identified triggers, airflow limitation, the role of comorbidities, symptom severity, response to treatment, and inflammatory biomarkers [47,48,49]. Although this classification appears useful, it doesn’t help to predict response to advanced therapies, such as biologics. Therefore, a better understanding of physio-pathological mechanisms underlying asthma could help to develop a personalized treatment, especially for severe asthma patterns [47].

Currently, two types of asthma endotype are recognized: (1) type-2 endotype and (2) non-type 2 endotype. Type 2 asthma is typical of atopic patients and is characterized by high sputum (>2%) and blood eosinophil counts (>300/µL), and high FeNO levels (>20 ppb) [47]. Interleukin (IL)-4, IL-5, and IL-13 (often produced after allergen exposure) and IL-33 [50], IL-25, and thymic stromal lymphopoietin (produced after the activation of the innate immune system by viruses and bacteria) [47] are involved in the inflammatory response.

Conversely, non-type 2 asthma is defined by neutrophilic and paucigranulocytic airway inflammatory patterns and a poor corticosteroid response [47]. Some studies support the hypothesis that elevated levels of circulating IL-17, IL-6, IL-23, bacterial infection, and obesity are all involved in the pathogenesis of the non-type 2 asthma [51].

### 5.1. Type 2 Asthma

Type 2 asthma covers more than 50% of asthma endotypes [52]. Type 2 asthma includes allergic (non)-eosinophilic, non-allergic eosinophilic, non-allergic non-eosinophilic, and mixed granulocytic phenotypes that often configure clinical pictures of difficult and severe asthma, for which monoclonal antibodies targeting type-2 inflammation appear to be the most promising emerging therapeutic strategies.

There are currently five monoclonal antibody therapies approved for severe asthma by the Food and Drug Administration (FDA): benralizumab, dupilumab, mepolizumab, omalizumab, and reslizumab [53], but only four of them are available for patients aged over 6 years [54] (Table 6).

Several considerations, including the patient’s age, degree of eosinophilia, IgE levels, presence of comorbidities, frequency, and route of administration, and delivery methods by the health system, must be considered when choosing a biologic treatment.

Benralizumab is a humanized Mab that binds to the α subunit of the IL-5 receptor (IL-5Rα), blocking the binding of IL-5 to its receptor and resulting in inhibition of eosinophil differentiation and maturation in the bone marrow. Furthermore, it promotes the apoptosis of both circulating and tissue-resident eosinophils. Its use has been approved as add-on maintenance treatment for adults and children (≥12 years) with severe eosinophilic asthma (baseline blood eosinophil cell counts >300 cells/μL or > 150 cells/μL for oral corticosteroids-dependent patients) despite proper treatment (high-dosage ICS + LABA). Studies report a good efficacy profile as a reduction in annual asthma exacerbations, significant improvement in prebronchodilator FEV_1,_ and steroid-sparing effect [55,56,57,58,59]. Adverse events like worsening of asthma and recurrent upper and lower respiratory tract infections are reported [60], but further short and long-term safety studies are to be implemented.

Dupilumab is a human IgG4 Mab that targets the IL-4 receptor alpha chain (IL-4Rα), blocking the production of IL-4 and IL-13 and the activation of eosinophilic inflammation [61,62]. Its efficacy is described in several atopic diseases like atopic dermatitis, asthma, and chronic rhinosinusitis with nasal polyposis [63,64]. Its use has been approved as an add-on maintenance treatment for adults and adolescents (≥12 years) with severe eosinophilic asthma, defined by raised blood eosinophils (>150 cells/μL) and/or raised FeNO (>20 ppb), and chronic rhinosinusitis with nasal polyps that are uncontrolled despite optimal treatment (medium-/high-dose ICS, plus up to 2 additional controllers including oral corticosteroid) [65]. It has been shown to reduce severe exacerbation, the use of oral corticosteroids (OCS) and rescue medication, to improve FEV_1_ and asthma control, and to suppress T2 inflammatory biomarkers in patients with uncontrolled, moderate-to-severe asthma with or without evidence of allergies [66]. The significant improvement of FEV_1_ from baseline to 12 weeks and 24 weeks is registered in the adult subgroup with blood eosinophils >300 cells/μL or with FeNO levels >50 ppb [65,67,68].

Although short-term safety data are reassuring, more accurate reports of adverse events are needed, in combination with long-term safety evaluation.

Mepolizumab is a humanized Mab (IgG1 kappa) directed against the IL-5 ligand that consequently can not interact with the IL-5 receptor, reducing the production and survival of eosinophils [69]. Its use has been approved for adults and children (≥6 years) affected by severe asthma and peripheral eosinophilia (blood eosinophil level of either 300 cells or more per μL in the past 12 months or 150 cells or more per μL at initiation). The recorded effects include the reduction of circulating eosinophils, exacerbations necessitating rescue medications, emergency room visits and hospitalizations, and OCS use, other than the improvement in asthma symptom scores and FEV_1_ from baseline [70,71,72,73,74,75]. Mild systemic symptoms (headaches, fatigue, arthralgia) and local injection-site reactions are the most common adverse events. As no drug-related anaphylaxis or fatal adverse events are reported, mepolizumab’s safety profile can be considered favorable [76,77,78,79].

Omalizumab is a recombinant humanized IgG1 monoclonal anti-IgE antibody that down-regulates the IgE high-affinity receptor (FcεRI) expression on basophils, mast cells, and dendritic cells (DCs), decreasing T2 cytokine production and inhibiting the eosinophilic inflammation. Its use is recommended in adults and children (≥6 years) with moderate-to-severe chronic allergic asthma, increase total IgE levels (total IgE level of 30–700 IU/mL considered in US; total IgE level of 30–1500 IU/mL considered in EU), and allergic sensitization to at least one perennial allergen [53]. Patients with higher levels of peripheral eosinophil count, exhaled nitric oxide (eNO), and serum periostin are more likely to respond to omalizumab [80,81]. Omalizumab provides clinically relevant improvements in exacerbation rate, lung function, and circulating eosinophil counts in children, adolescents and adults with moderate-to-severe uncontrolled asthma [82,83]. Mild systemic (pyrexia, headache, abdominal pain) and local (swelling, erythema, pain, pruritus in the injection side) adverse reactions are reported [84]. Based on recent literature, both skin and systemic adverse reactions are due to immune complexes formed between omalizumab and IgE [85]. As life-threatening systemic reactions such as anaphylaxis are rarely reported [86], the FDA added a black box warning, so that physicians administering omalizumab should be able to manage anaphylaxis, and patients should be monitored in a safe setting after administration.

Allergen-specific immunotherapy (AIT) is the only causal treatment in allergic asthma, but it is exclusively suggested in mild-moderate asthma. As it can induce asthma exacerbation, it cannot be prescribed in patients with uncontrolled or severe asthma [33].

### 5.2. Non-Type 2 Asthma

For patients with neutrophilic asthma, only a few strategies are available and include long-acting anticholinergic bronchodilator, theophylline, and macrolides.

Tiotropium is a long-acting anticholinergic bronchodilator approved by the FDA for ages 6 years and older. It appears useful in patients with severe asthma at risk of β2-receptor downregulation due to overuse of short-acting β2-agonists [87,88].

Theophylline is a molecule with bronchodilator and anti-inflammatory effects [89,90] able to improve steroid sensitivity [90]. As it promotes neutrophil apoptosis, it could be beneficial to patients with neutrophilic asthma. The side effects (nausea, vomiting, headache, irritability, insomnia, tremors) and the need for frequent drug level monitoring limits its use in routine practice. However, its administration is associated to a reduction in daily oral corticosteroid use and improvement in FEV1 [90].

Macrolides are not currently recommended in severe pediatric asthma, but the prescription of low-dose macrolide as add-on therapy may be considered in children with refractory disease, oral corticosteroid dependence, eosinophilic or non-eosinophilic inflammation, and recurrent lower respiratory tract infections [88].

## 6. Conclusions

Asthma is a common disease in childhood, with a minority of affected children suffering from severe asthma. Uncontrolled severe asthma represents a challenge for physicians; therefore, a multidisciplinary systematic assessment is warranted. Early identification of modifiable factors for children with difficult-to-treat asthma allows establishing better control of asthma without the need for further invasive investigations and treatment escalation. Otherwise, addressing a correct diagnosis of true severe therapy-resistant asthma avoids diagnostic and therapeutic delays. Once the patient with severe asthma has been identified, the definition of clinical phenotype and endotype could help the clinician to resort to a “personalized medicine”, which recently also includes new biological drugs. Unfortunately, these novel therapies are not available in preschool age, hence, the control of modifiable factors and the empowerment of parents and caregivers remain crucial for the management of patients belonging to this age group.

## Figures and Tables

**Table 1 children-07-00286-t001:** From symptoms to diagnosis: alternative diagnosis mimicking asthma.

Symptoms	Diseases	Investigations
Stridor at rest Stridor on exertion Dyspnea on exertion	Congenital/acquired subglottic stenosis Vocal cord dysfunction	Functional respiratory test Flexible fibreoptic rhino-laryngoscopy
Productive cough Recurrent wheezing Failure to thrive	Cystic fibrosis Non-CF bronchiectasis Churg Strauss syndrome Tracheo-esophageal fistula Aspiration	Sweat test Serum eosinophils count ANCA Exhaled nasal NO Nasal brushing Chest CT scan Flexible bronchoscopy Lipid laden alveolar macrophage on BAL
Dry cough/noisy breathing Unresponsive to SABA Dyspnea on exertion	Tracheo-bronchomalacia Vascular ring	Functional respiratory tests Flexible bronchoscopy Chest CT scan with contrast enhancement and dynamic study
Symptoms onset from birth Drumstick fingers	Congenital lung disease Heart diseases	Oximetry Chest X-ray Chest CT scan Heart ultrasound
Heartburn Chest pain	Gastroesophageal reflux	pH-impedance

**Table 2 children-07-00286-t002:** Tools for monitoring the adherence to prescribed treatment.

Measurement Tools of Treatment Adherence	
Subjective	Physician assessment of adherence
	Self-report questionnaires
	Morisky scale Medication adherence report scale
Objective	Drug levels
	Exhaled nitric oxide
	Prescription data
	Weighing inhaler canisters
	Dose counters
	Directly observed therapy
	Nurse home visits
	Electronic monitoring devices
	Integrating digital technologies

**Table 3 children-07-00286-t003:** Asthma comorbidities.

Comorbidity	Diagnosis	Treatment
Rhinosinusitis/nasal polyposis	ENT evaluation Sinus CT	Topic CS Surgery
Allergic rhinoconjuctivitis	Anamnestic data SPT test Specific IgEs	Allergen avoidance Topic CS Antihistamines Antileukotriene
Dysfunctional breathing	Anamnestic data Nijmegen questionnaire	Breathing rehabilitation
Vocal cord dysfunction	ENT evaluation Laryngoscopy	Speech retraining
Obesity	BMI	Diet
Obstructive sleep apnea	Anamnestic data Polisomnography	Weight loss Nocturnal CPAP
Gastroesophageal reflux	PPI trial pH-impedance	PPI Lifestyle changes
Bronchiectasis	Chest CT scan	Hypertonic solutions Physiotherapy Macrolide
Bronchopulmonary aspergillosis	Total IgE IgE for Aspergillus Fumigatus IgG for Aspergillus Fumigatus Chest CT scan	Prednisone Voriconazole

**Table 4 children-07-00286-t004:** High-dose ICD dosages for children (6–11 years), adolescents, and adults (mcg/d) according to Global Initiative for Asthma (GINA 2020) guidelines.

**Adults and Adolescents (12 Years and Older)**	
**Inhaled Corticosteroid**	**Total Daily ICS Dose (mcg)**
Beclomethasone dipropionate (pMDI, standard particle, HFA)	>1000
Beclomethasone dipropionate (pMDI, extrafine particle, HFA)	>400
Budesonide (DPI)	>800
Cilesonide (pMDI, extrafine particle, HFA)	>320
Fluticasone furoate (DPI)	200
Fluticasone propionate (DPI)	>500
Fluticasone propionate (pMDI, standard particle, HFA)	>500
Mometasone furoate (DPI)	400
Mometasone furoate (pMDI, standard particle, HFA)	>400
**Children 6–11 Years**	
**Inhaled Corticosteroid**	**Total Daily ICS Dose (mcg)**
Beclomethasone dipropionate (pMDI, standard particle, HFA)	>400
Beclomethasone dipropionate (pMDI, extrafine particle, HFA)	>200
Budesonide (DPI)	>400
Budesonide (nebules)	>1000
Cilesonide (pMDI, extrafine particle, HFA)	>160
Fluticasone furoate (DPI)	n.a.
Fluticasone propionate (DPI)	>200
Fluticasone propionate (pMDI, standard particle, HFA)	>200
Mometasone furoate (pMDI, standard particle, HFA)	200

pMDI, pressurized metered-dose inhaler; HFA, hydrofluoroalkane propellant; DPI, dry powder inhaler; n.a, not available.

**Table 5 children-07-00286-t005:** Low total daily dose (mcg) of ICS for children 5 years and younger, according to the Global Initiative for Asthma (GINA 2020) guidelines.

Inhaled Corticosteroid	Low Total Daily Dose (mcg)
Beclomethasone dipropionate (pMDI, standard particle, HFA)	100
Beclomethasone dipropionate (pMDI, extra-fine particle, HFA)	50
Budesonide (nebules)	500
Fluticasone propionate (pMDI, standard particle, HFA)	50
Mometasone furoate (pMDI, standard particle, HFA)	100

pMDI, pressurized metered-dose inhaler; HFA, hydrofluoroalkane propellant; DPI, dry powder inhaler.

**Table 6 children-07-00286-t006:** FDA and EMA approved biologic drugs for severe pediatric asthma.

Biologics	Mechanism of Action	Indications	Dosing Route
Benralizumab	Anti-IL5 (binds to IL5 receptor; causes apoptosis of eosinophils and basophils)	≥2 ys old Add-on maintenance treatment for adults and children with severe eosinophilic asthma (baseline blood eosinophil cell counts >300 cells/μL or >150 cells/μL for OCS-dependent patients) despite high-dosage ICS + LABA	30 mg s.c. every 4 wk for 3 doses; then every 8 wk
Dupilumab	Anti-IL4 (binds to IL4 receptor; blocks activation of IL4 and IL13 mediated inflammation)	≥12 ys old Add-on maintenance treatment for adults and adolescents with severe eosinophilic asthma (blood eosinophils > 150 cells/μL and/or raised FeNO >20 ppb) and chronic rhinosinusitis with nasal polyps which are uncontrolled despite medium-/high-dose ICS plus up to 2 additional controllers including OCS	200 or 300 mg s.c. every 2 wk
Mepolizumab	Anti-IL5 (binds to IL5 ligand; prevents IL5 from binding to receptor)	≥6 ys old Treatment for adults and children affected with severe asthma and peripheral eosinophilia (blood eosinophil level of either 300 cells or more per μL in the past 12 months or 150 cells or more per μL at initiation)	100 mg s.c. every 4 wk
Omalizumab	Anti-IgE (prevents IgE from binding to receptor)	≥6 ys old Treatment for adults and children with moderate-to-severe chronic allergic asthma, increase total IgE levels (total IgE level of 30–700 IU/mL considered in US; total IgE level of 30–1500 IU/mL considered in EU) and allergic sensitization to at least one perennial allergen	30–1500 IU/mL s.c. every 2–4 wk

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
