# Peer review of "Difficult and Severe Asthma in Children"

_children, 2020, doi:10.3390/children7120286_

Round 1

Reviewer 1 Report

This is an interesting review with the aim to define and differentiate between difficult-to-treat and true severe therapy-resistant asthma.

The flow of the paper is illogical and in need of improvement in order to guide the reader through the review and be convincing in delivering its main message to the reader.

After reading the paper, peers will know what severe therapy-resistant asthma is, but have not really been helped in facing difficult-to-threat asthma.

Sub-headlines to distinguish major from less major paragraphs could be of some help but a revision is still needed to improve the flow.

Below are some suggestions/comments for the authors.

  1. It would be beneficial if difficult-to-treat asthma was rephrased as difficult asthma in one of the leading paragraphs in order to remove confusion and intermingling these exchangable definitions throughout the manuscript.
  2. Are respiratory resistance and reactance (pp. 2 line 67) used to diagnose asthma? If yes, in what manner?
  3. More thorough rationales for listing the symptoms in Table 1 would clearly benefit the manuscript. How often are the listed symptoms used to suspect an asthma diagnosis? Are failure to thrive and drumstick fingers really asthma symptoms?
  4. I find that difficult/difficult-to-treat asthma is vaguely defined in that the paragraph with the headline ”Difficult asthma” pp. 3 line 78-85 does not define nor mention difficult-to-treat asthma.
  5. Explanations of abbreviations are lacking throughout the manuscript. Consider making a list.
  6. Numbers for headlines on pp.´s 3, 4 and 5 are not incrementing: 1). Adherence to medication, 1). Environmental exposures etc. Remove punctuations in these so e.g. the first becomes ”1) Adherence to medication”.
  7. The tables 4 and 5 with doses of inhaled corticosteroid seems kind of non-necessary. A reference would be sufficient. When browsing the GINA pocket guide through the weblink I was unable to see dose guidelines for children 5 yrs and younger.
  8. When mentioning type 2 asthma, headline pp.7 line 182, it is unclear whether this asthma endotype is found and to what extent in controlled asthma, difficult asthma and severe therapy resistant asthma, respectively.
  9. Please define paediatric age, pp.7 line 187.

And with regards to language, grammar, and formatting, a few extra issues:

  1. Line 23-23, pp. 1, please rephrase: ”…being responsible of the greatest social and economic burdens (2). …”.
  2. Line 47, pp. 2, increase in ”…, increase disease burden, …” should be increased.
  3. Line 98, pp.3, please add and after the semicolon in the sentence ”…to manipulate the data; …”.
  4. Consider turning the row/columns around in Table 2: Make two table rows, one with subjective and the next with objective. This will ease reading of the table.
  5. Line 118, pp. 4, add it after the comma in the context ”…(32). Consequently, …”.
  6. Line 145, pp. 5, long-acting beta2 agonist, please write with a β as on pp. 1 or use the LABA abbreviation.
  7. Line 170-71 pp. 6. Please rephrase the sentence ”Therefore, research focused the patho-mechanisms involved in asthma in order to develop a personalized treatment for severe asthma patterns (45).”.
  8. Line 194, pp. 7. Swap both and of in the sentence ”…promotes the apoptosis both of circulating…”.

Author Response

Dear Reviewer,

Thanks for your critical comments.

We adjusted the manuscript to the requests of all the reviewers.

As the Reviewer number 2 and 3 appreciated the structure of the manuscript, we didn’t modify it, but we added the information requested by you. We hope to have satisfied your questions.

We attacched the file with a point by point response. 

Reviewer 2 Report

The manuscript is well written and no major concern regarding this manuscript was detected.

1. The manuscript covers from basic to extended medication, such as biologics, regarding severe asthma.
2. The manuscript covers recent publication to support their view.

Author Response

Dear reviewer,

Thanks for your positive comments and appreciation.

Reviewer 3 Report

This manuscript is a much-needed review on the subject of severe and difficult to treat asthma in children. Phenotypic heterogeneity, medication adherence and appropriate environmental assessments and interventions are important considerations in the management of severe childhood asthma. I compliment the authors for their work.

If I may, I would suggest you include the role of socioeconomic factors in the review, as children in low-income neighbourhoods are particularly vulnerable to asthma and those with socio-economically disadvantaged backgrounds are more prone to die from asthma.

I would also suggest that you stress further that not many therapeutic approaches exist for children with severe asthma of pre-school age, hence the crucial importance of environmental control and literacy of the parents and caregivers.

Author Response

Dear Reviewer, 

Thanks for your positive comments.We added information as suggested.  

Round 2

Reviewer 1 Report

The revised paper is acceptable for publication.